# Flexible Prefrontal Control over Hippocampal Episodic Memory for Goal-Directed Generalization

**Yicong Zheng**[1,2,3], **Nora Wolf**[1,2], **Charan Ranganath**[1,2], **Randall C. O'Reilly**[1,2,3,4], **Kevin L. McKee**[3]

[1]Department of Psychology, University of California, Davis, California, United States of America

[2]Center for Neuroscience, University of California, Davis, California, United States of America

[3]Astera Institute

[4]Department of Computer Science, University of California, Davis, California, United States of America

## Abstract

**Many tasks require flexibly modifying perception and behavior based on current goals. Humans can retrieve episodic memories from days to years ago, using them to contextualize and generalize behaviors across novel but structurally related situations. The brain's ability to control episodic memories based on task demands is often attributed to interactions between the prefrontal cortex (PFC) and hippocampus (HPC). We propose a reinforcement learning model that incorporates a PFC-HPC interaction mechanism for goal-directed generalization. In our model, the PFC learns to generate query-key representations to encode and retrieve goal-relevant episodic memories, modulating HPC memories top-down based on current task demands. Moreover, the PFC adapts its encoding and retrieval strategies dynamically when faced with multiple goals presented in a blocked, rather than interleaved, manner. Our results show that: (1) combining working memory with selectively retrieved episodic memory allows transfer of decisions among similar environments or situations, (2) top-down control from PFC over HPC improves learning of arbitrary structural associations between events for generalization to novel environments compared to a bottom-up sensory-driven approach, and (3) the PFC encodes generalizable representations during both encoding and retrieval of goal-relevant memories, whereas the HPC exhibits event-specific representations. Together, these findings highlight the importance of goal-directed prefrontal control over hippocampal episodic memory for decision-making in novel situations and suggest a computational mechanism by which PFC-HPC interactions enable flexible behavior.**

**Keywords:** prefrontal cortex; hippocampus; reinforcement learning; episodic memory; generalization

## Introduction

A fundamental aspect of intelligence is the ability to learn from experience and apply that knowledge to guide future decisions. While artificial intelligence (AI) systems can effectively utilize training data for some generalization within their learned distribution, they struggle with out-of-distribution generalization due to their design focus on capturing statistical patterns. In contrast, biological systems demonstrate remarkable flexibility in adapting to novel situations in a few shots through their capacity to leverage both episodic memories of specific events and semantic knowledge accumulated through experience. Humans can even transfer knowledge across seemingly unrelated situations when they share underlying principles - a phenomenon known as "far transfer" (as opposed to "near transfer" where situations might appear more similar) (Barnett & Ceci, 2002). A key question then is how different brain regions might work in concert to generalize knowledge quickly and adaptively, and how this can flexibly guide behavior.

Far transfer relies on two critical components: 1) knowledge about the past, and 2) the ability to associate the current situation with the past. The hippocampus (HPC) has been thought of as being critical for storing context-specific episodic memories, as well as matching current contexts with past experiences. Motivated by the brain's efficient handling of episodic memory, recent AI research has sought to mimic these processes. Approaches such as kNN-LM (Khandelwal et al., 2020), which combines a pretrained language model with a k-nearest neighbor search in a large memory bank, and Retrieval Augmented Generation (RAG) (Lewis et al., 2021), where a transformer queries an explicit memory bank for semantically relevant information, represent early steps toward integrating episodic-memory-like systems in AI. More recently, researchers found that combining brain-inspired memory systems akin to short-term and long-term memory (including semantic and episodic memory) with transformers boost model performance in various domains beyond language modeling (Behrouz et al., 2024).

To use episodic memory effectively, particular memories must be selected from a very large set to include information that is maximally relevant to the current context. A simple heuristic is to recall memories from previous situations that presented similar sensory stimuli. However, far transfer or analogical reasoning is better defined by the perception and utilization of unobvious structural similarities among situations. Mounting evidence suggests that the prefrontal cortex (PFC) is essential for selecting and integrating the right memories for generalization, exerting top-down control over HPC memory processing (Eichenbaum, 2017). Specifically, the PFC is thought to support selective processing and maintenance of task-relevant information and guide goal-directed memory retrieval (Eichenbaum, 2017), and actively integrate new environmental cues with established represen-

tations from the HPC to form and update its working memory state (Miller & Cohen, 2001; Ranganath, 2010; Eichenbaum, 2017; Preston & Eichenbaum, 2013). Thus, mammalian species may generalize to novel situations better through interactions between the HPC and the PFC, whereas current AI research has been lacking a prefrontal-like mechanism (Russin et al., 2020; LeCun, 2022).

Damage to the PFC impairs an organism's ability to adapt to novel situations by disrupting the integration of new information into existing knowledge frameworks (Eichenbaum, 2017). Lesions in the PFC hinder learning overlapping stimulus pairs, making transitive inferences, and distinguishing contexts (DeVito et al., 2010; Xu & Südhof, 2013). PFC damage also affects spatial learning and the ability to adapt strategies in dynamic environments, as evidenced by deficits in tasks like the Morris water maze under changing conditions (Mogensen et al., 1995; Compton et al., 1997; de Bruin et al., 1994; Lacroix et al., 2002). Beyond encoding, the PFC is crucial for effective memory retrieval, selecting contextually appropriate memories, and suppressing irrelevant representations to resolve competition between similar memories (Preston & Eichenbaum, 2013). Finally, PFC is at the core of analogical reasoning (Hobeika et al., 2016; Whitaker et al., 2018), a key component for enabling far transfer. These findings collectively highlight the PFC's essential role in flexibly adapting to new situations that may have a different context than past experiences.

Despite advances in understanding how HPC and PFC could interact to achieve generalization to similar or dramatically different contexts, models have yet to fully capture the dynamic interplay between these regions that supports generalization to a novel situation. Moreover, existing models do not provide a clear computational mechanism by which the HPC and PFC can learn to perform complex decision making tasks in an integrated end-to-end system. Here, we propose a reinforcement learning agent architecture that combines PFC working memory, implemented as an RNN (Wang et al., 2018), with HPC episodic memory stored in a key-value system (Gershman et al., 2025) queried by either PFC modulated input, or raw sensory input during memory retrieval. In our model, the PFC learns to encode and retrieve episodic memories from the key-value buffer and uses self-attention to integrate past experiences with its current state to guide action selection. The agent continuously explores and exploits a series of continuously generated environments simulating a Morris water maze task, learning to use episodic memory during exploration to improve decision-making during exploitation in a few-shots through effective meta-learning (Wang et al., 2018). We present three experiments demonstrating that: (1) similarities in sensory input can be used to effectively recall episodic memory for guiding current decision making, (2) additional steps of processing that imitate the role of the PFC can be trained to transfer memory based on structural relations, rather than sensory similarity, (3) the PFC can learn to flexibly control which particular structural relations are used to

invoke episodic memories given changes in the goal context.

## Methods

### Reinforcement Learning Environment

We created a reinforcement learning environment that continuously generates new mazes for the agent to explore, and samples from a set of previously explored environments for the agent to exploit.

**Base Environment**  We introduce an episodic Morris water maze environment in discrete grid-world settings, formulated as a partially observable Markov decision process (POMDP). The agent observes a local $3 \times 3$ grid and a continuous "context" vector that belongs to a maze. Each maze is instantiated on a configurable $4 \times 4$ grid, where a target and an agent are placed in distinct locations.

The environment contains two trial types, determined probabilistically for each episode: explore trials, where a new maze is generated with an updated context vector, and exploit trials, where a maze is sampled from a history of 5 previous mazes. The trial type is determined according to:

$$trialtype = \begin{cases} \text{explore,} & \text{with probability } p \\ \text{exploit,} & \text{with probability } 1 - p \end{cases} \quad (1)$$

where $p$ = 50%. The trial type remains constant throughout each episode, and is re-sampled at the start of each new episode. During explore episodes, a new maze is generated with a new random seed and remains fixed for all 5 trials within that episode. For exploit episodes, each trial's maze is independently sampled from a history of 5 previously encountered environments.

At each time step, the agent receives an observation composed of a flattened $3 \times 3$ subgrid centered on its current position and a context vector that provides contextual information about the maze. The observation space can be formally defined as:

$$\mathbf{obs}_t = [\mathbf{v}_t, \mathbf{e}] \quad (2)$$

where $\mathbf{v}_t$ is the sensory input (the $3 \times 3$ subgrid) and $\mathbf{e}$ is the fixed maze context vector. Note that the context vector $\mathbf{e}$ remains constant throughout exploration of a given maze.

The action space includes four discrete moves (up, down, left, right). Movement is constrained by grid boundaries, ensuring that the agent remains within the maze. Each action incurs a small penalty, whereas reaching the target yields a positive reward. Trials terminate when the target is reached or the step limit is exceeded.

**Asymmetrical Environment**  To test generalization beyond surface similarity (as what is relevant is not always similar, and what is similar is not always relevant), we introduce a variant of the episodic water maze environment that includes an asymmetrical tagging mechanism. In asymmetrical episodic water mazes, structurally related mazes (i.e., mazes with the same hidden platform location for reward but different contexts) during explore and exploit trials are tagged with different context

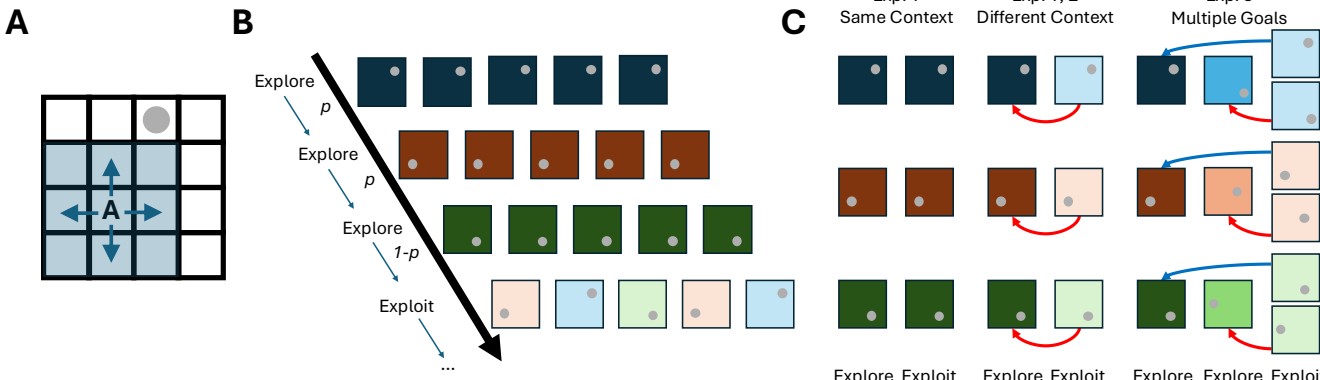

Figure 1: Task structure. **A.** Simulated $4 \times 4$ Morris water maze for reinforcement learning. An agent starts at a random location in the maze, seeking a hidden platform for reward with a slight penalty for each action. The action space consists of four possible moves: up, down, left, and right. The observation space of the agent at a given moment contains a $3 \times 3$ subgrid centered on its current position and a random context vector for the current maze. Note that only walls and the agent itself are visible. **B.** Task procedure. Each row represents one episode, which contains 5 trials of a new maze during explore, or 5 trials of mazes sampled from a history of mazes during exploit. Episode type is sampled by probability $p$. **C.** Task variants. The context vector for explore and exploit trials can be either the same (as in Exp. 1), or different (as in Exp. 1 and 2), where the explore context is a transformed version of the exploit context to represent arbitrary structural relationships between them (indicated by the red arrow). In Exp. 3, the two mazes for explore trials contain context vectors that are transformed versions of the context vector for exploit trials using two predefined transformation matrices across all mazes (indicated by the red and blue arrows). An additional goal bit is provided in the observation to indicate the current goal reward location during exploit trials.

vectors with a fixed relationship between them:

$$\mathbf{e} = \begin{cases} \mathbf{W} \cdot \mathbf{e_{base}}, & \text{explore trials} \\ \mathbf{e_{base}}, & \text{exploit trials} \end{cases} \quad (3)$$

where $\mathbf{W}$ is a fixed transformation matrix and $\mathbf{e_{base}}$ is the base context vector for the maze.

This ensures the agent never encounters the base context during explore trials, requiring learned mappings to retrieve episodic memories for decision-making.

**Asymmetrical Environment with Multiple Goals**  To test generalization with multiple goals, we extend the asymmetrical environment to include mazes with multiple goals during exploit trials. A 1-bit goal indicator tells the agent which goal to pursue. The agent must learn to retrieve memories of previously seen mazes relevant to the current maze and goal. Each goal corresponds to a unique context vector transformation for explore trials, while exploit trials use the base context vector:

$$\mathbf{e} = \begin{cases} \mathbf{W_1} \cdot \mathbf{e_{base}}, & \text{explore trials (goal 1)} \\ \mathbf{W_2} \cdot \mathbf{e_{base}}, & \text{explore trials (goal 2)} \\ \mathbf{e_{base}}, & \text{exploit trials} \end{cases} \quad (4)$$

The observation space extends to include the goal bit:

$$\mathbf{obs}_t = [\mathbf{v}_t, \mathbf{e}, \mathbf{g}] \quad (5)$$

where $\mathbf{g}$ is a binary indicator for the current goal.

In explore episodes, the agent learns about two different goals in separate trials, where each goal has a unique context

vector derived from the same base vector. Explore trials can be either blocked or interleaved. In exploit episodes, the agent must retrieve previously seen, structurally related mazes with the same goal as the current goal, demonstrating its ability to utilize learned knowledge.

### Agent Architecture

The agent mimics brain-like working and episodic memory integration for decision-making. Its architecture includes:

1. A hippocampal (HPC) module that stores and retrieves episodic memories using a query-key-value architecture (detailed below)

2. A PFC module that:

   (a) Uses a recurrent neural network (RNN), implemented as a reservoir network (Lukoševičius & Jaeger, 2009), to integrate past rewards, actions, and observations into working memory

   (b) Learns to interact with the HPC for episodic control

   (c) Uses a self-attention mechanism to integrate episodic memory with working memory

**Base Agent**  The agent was designed to both learn memory tasks as efficiently as possible and to reflect plausible natural mechanisms of short-term memory. It has been found that Echo State Networks (ESNs Jaeger, 2007), a type of reservoir computing (Lukoševičius & Jaeger, 2009), outperform gated memory architectures on tasks that involve meta-learning and

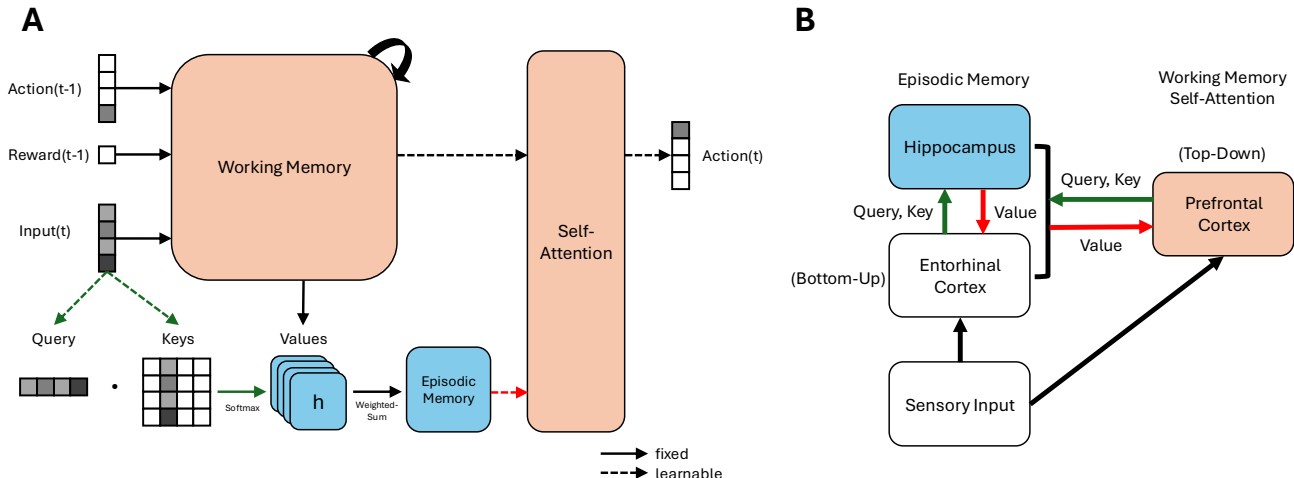

Figure 2: Overview of the agent architecture. **A.** The agent contains a RNN as its working memory for keeping beliefs about the parts of the environment that are not currently observable, as well as its historical actions and corresponding rewards given state inputs. It also contains a q-k-v system for episodic memory, which can be attended over by working memory in a self-attention layer before making the final output action. **B.** Potential functional mapping onto anatomical regions and pathways in the brain. The bidirectional pathways between PFC and HPC (including indirect pathways through the entorhinal cortex) can be used both during encoding and retrieval for PFC top-down modulation of HPC activities. Hippocampal memories are stored as key-value pairs, where key represents a function of the environmental input, and value represents the hidden state of the RNN at the time of encoding. During retrieval, the PFC (top-down) or the entorhinal cortex (bottom-up) can send in queries for retrieving the most relevant memory, represented as a weighted sum of the values based on the similarity between the query and keys. See Supplementary Materials for diagrams for encoding and retrieval.

non-Markovian time dependence (McKee, 2024). The addition of reward-driven input filtering greatly improves training time when there the task involves many input dimensions that do not usefully condition the agent's policy, and resembles the concept of selective attention in natural intelligence (McKee, 2025). Hence, our agent used an ESN (with hyperparameters from McKee (2024)) for short-term memory with reward-driven input filtering. At each timestep $t$, the RNN input is the concatenation of the previous reward $r_{t-1}$, previous action $a_{t-1}$ (one-hot), and current observation $obs_t$, modulated by a learned filter signal $m$:

$$x_t = [r_{t-1}, a_{t-1}, obs_t] \cdot m \tag{6}$$

$$h_t = \text{RNN}(x_t, h_{t-1}) \tag{7}$$

The modulation signal $m$ is generated by a neural network that takes a fixed, high-dimensional bias vector $b$ and outputs values bounded between $m_{\min}$ and $m_{\max}$:

$$m = (m_{\max} - m_{\min}) \cdot \text{sigmoid}(f_{\text{filter}}(b)) + m_{\min} \tag{8}$$

The agent retrieves relevant episodic memories and integrates them with its current hidden state through a self-attention mechanism. The resulting context-enriched representation is then passed through a Multilayer Perceptron (MLP) to compute Q-values:

$$Q(s_t, a) = f_{\text{MLP}}(\text{Attn}(h_t, \text{EM})) \tag{9}$$

where Attn represents the self-attention mechanism, and EM represents retrieved episodic memories (detailed in sections below).

Learning uses double Q-learning with Huber loss and filter regularization:

$$\mathcal{L} = \mathbb{E}\left[ \text{Huber}\left( Q_\theta(s_t, a_t) - y_t \right) \right] + \lambda_{\text{filter}} \, \mathbb{E}[m] \tag{10}$$

$$y_t = r_t + \gamma Q_{\theta^-}\left( s_{t+1}, \arg\max_{a'} Q_\theta(s_{t+1}, a') \right) \tag{11}$$

where $y_t$ is the target Q-value from Double Q-learning (van Hasselt et al., 2016). $\theta$ represents the online network parameters, and $\theta^-$ represents the target network. The discount factor $\gamma$ determines the relative importance of future rewards. The Huber loss minimizes the Bellman error to maximize the rewards obtained by sampling actions from the policy. Finally, the filter regularization term $\lambda_{\text{filter}} \, \mathbb{E}[m]$ pressures inputs to scale toward zero unless counteracted by backpropagated gradients of the Bellman error, resulting in suppression of unnecessary information in the RNN state early in training.

**Query-Key-Value Memory in the Brain** The episodic memory module is implemented as a key–value buffer for simplicity while maintaining key principles in HPC episodic memory (Gershman et al., 2025). At the end of each trial (event), the agent stores key–value pairs, following insights from models storing episodic memory at event boundaries (Lu et al., 2022):

$$q = f_q(x), \quad k = f_k(x), \quad v_{\text{encoded}} = h \tag{12}$$

4

where both $f_q$, the query function, and $f_k$, the key function, are identity functions in Experiment 1, and parameterized as MLPs in Experiments 2 and 3. $v_{\text{encoded}}$ is just the reservoir hidden state $h_t$ at the time of encoding, which contains all relevant information about the event.

Memory retrieval is performed via a softmax-weighted sum over stored values, as $v_{\text{retrieved}} = \text{softmax}(\langle Q,K \rangle)V$, where $\langle Q,K \rangle$ represents a vector of query-key similarities (dot products), and the retrieved value $v_{\text{retrieved}}$ is a weighted sum of stored values (Hassabis & Maguire, 2009).

**Attention over Working Memory and Episodic Memory**
Our agent employs a self-attention mechanism to integrate retrieved episodic memories with the current working memory state for decision-making. The current working memory state and the retrieved episodic memory are first embedded into a common representation space through a shared MLP embedding network:

$$e_{\text{wm}} = f_{\text{emb}}(h_t), \quad e_{\text{em}} = f_{\text{emb}}(v_{\text{retrieved}}) \qquad (13)$$

The attention mechanism then uses the embedded current working memory state as the query to attend over itself and the embedded retrieved episodic memory:

$$\tilde{h}_t = \text{Attn}(e_{\text{wm}}, [e_{\text{em}}; e_{\text{wm}}], [e_{\text{em}}; e_{\text{wm}}]) \qquad (14)$$

$$\text{Attn}(q, K, V) = V \cdot \text{softmax}\left(\frac{qK^\top}{\sqrt{d_k}}\right) \qquad (15)$$

where $h_t$ represents the current RNN hidden state, $v_{\text{retrieved}}$ is the retrieved episodic memory value, $f_{\text{emb}}$ denotes the shared MLP embedding network, and $[e_{\text{em}}; e_{\text{wm}}]$ indicates the concatenation of embedded memories.

This attention mechanism allows the agent to dynamically weight the relevance of both retrieved episodic memories and recent working memory states when making decisions. The attended representation $\tilde{h}_t$ is then used to compute Q-values for final action selection using Eq. 9.

## Results

### Experiment 1: Episodic Memory Increases Exploitation Efficiency when Encoding-Retrieval Contexts are Similar

Reinforcement learning often struggles with quickly adapting to new situations. Theories suggest that episodic memory could be crucial in addressing this challenge (Gershman & Daw, 2017). However, episodic memory retrieval can be difficult in complex environments, especially when they are partially observable. We developed an episodic memory retrieval mechanism based on the Encoding Specificity Principle (Tulving & Thomson, 1973), which posits that episodic memory is more effectively retrieved when the retrieval context closely matches the encoding context. We tested this principle in our agent (Fig. 2) by comparing its performance in the Morris water maze task during exploit trials with context cues that were either similar or dissimilar to those in the explore trials (Fig. 1).

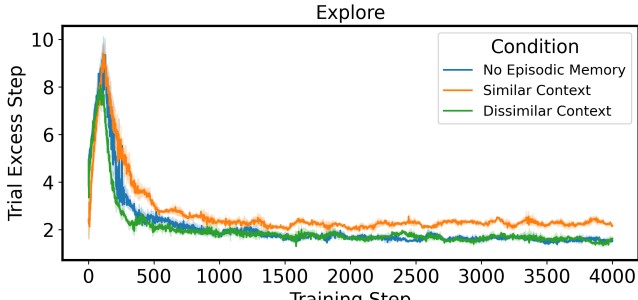

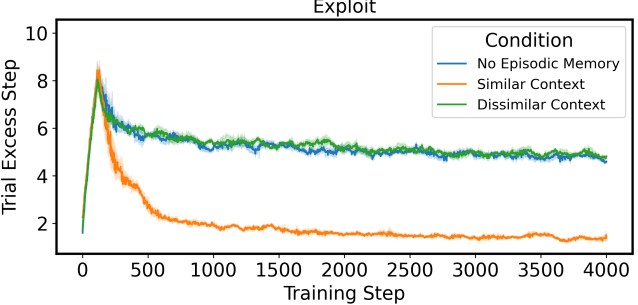

Figure 3: Experiment 1 results. Excess steps are the extra steps taken beyond the shortest path to the hidden target. The agent required fewer steps during exploit trials when the context cue matched that of the explore trials compared to when it differed. On the other hand, when exploit and explore context cues were dissimilar or when the agent had no episodic memory, performance was better during exploration than when contexts were similar and episodic memory could be used. NOTE: Shaded areas show SEM across runs.

In this task, the agent's objective was to learn to locate a hidden target location in a maze, with the maze identified by a context vector within the observation space, starting from random locations. Initially, the agent explores a new maze in each episode. Subsequently, with probability $p$, it enters either another *explore* episode that presents a new maze or an *exploit* episode that presents a previous maze randomly sampled from the task history. The key assumption is that if the context cues are similar across explore and exploit trials for the structurally related mazes (i.e., those with the same target location), the agent should be able to remember the explore trials and navigate directly to the hidden target location. On the other hand, if the context cues are dissimilar across functionally related pairs of explore and exploit trials, then the agent will not be able to exploit its memory based on cue similarity alone. Without the right memory selection strategy, episodic memory becomes only a source of noise that must be ignored when taking actions during both explore and exploit trials. This is further supported by an additional manipulation we tested (see Supplementary Materials) in which we added a gating mechanism that suppresses memory retrieval when the available memories are irrelevant.

Our results demonstrate that the agent required signifi-

cantly fewer steps to reach the hidden platform during exploit trials when the context cue matched that of the explore trials compared to when it differed (Fig. 3). This provides evidence that retrieving relevant episodic memories and integrating them with current state information (stored in working memory) can substantially improve exploitation efficiency. Interestingly, we found that agents without episodic memory or with dissimilar context cues performed better during exploration (Fig. 3). This suggests that learning to use episodic memory for some tasks can sometimes interfere with learning in novel environments, as the agent may inappropriately weight irrelevant memories when no truly relevant experiences exist (as during exploration episodes, the maze is presented is always a novel one). While counterintuitive, this finding aligns with the well-documented phenomenon of proactive interference (Jonides & Nee, 2006), where existing memories can impair the acquisition of new associations to similar stimuli. This highlights an important trade-off between the benefits of episodic memory for exploitation and its potential costs during learning of novel information.

## Experiment 2: PFC Top-Down Modulates Hippocampal Episodic Memory for Structure Learning

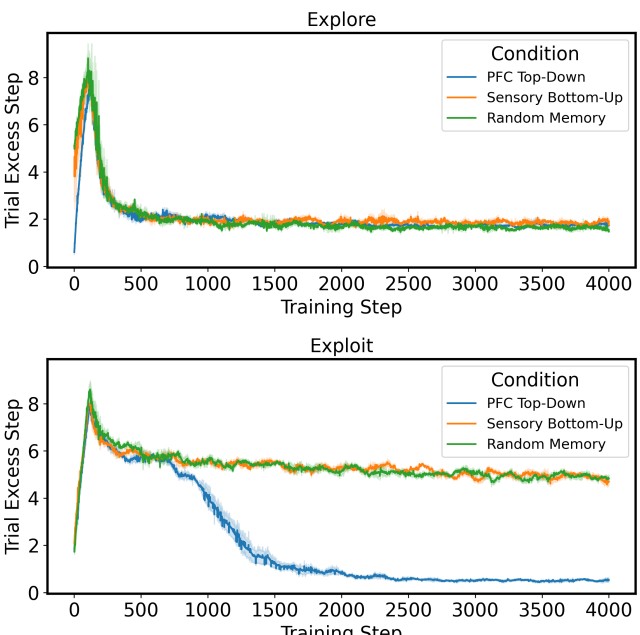

Figure 4: Experiment 2 results. The agent with PFC top-down modulation over hippocampal episodic memory significantly outperformed the sensory-driven bottom-up memory retrieval condition and the random memory condition during exploit. NOTE: Shaded areas show SEM across runs.

The results from Experiment 1 show that retrieving episodic memories is beneficial for decision making in environments that share the same context cues. However, in many real-world scenarios, the current situation may look dramatically different from past experiences, even though they are inherently related in an abstract way. For example, students often encounter math puzzles in school that are structurally similar to those they have solved before, but appears seemingly different on the surface. Students are known to have a difficult time transferring their knowledge to seemingly unrelated problems. This is commonly referred to as a phenomenon called "far transfer" (Barnett & Ceci, 2002), and is known to be more successful whenever the agent deeply grasps the underlying problem structure (Duncker & Lees, 1945; Gick & Holyoak, 1980, 1983). Moreover, neuroimaging studies show that individual differences in connectivity between HPC and PFC are correlated with far transfer (Gerraty et al., 2014). Here, we propose that PFC top-down modulates hippocampal activities during encoding and retrieval of episodic memories to enable learning of structures that are independent of the sensory input.

To test this hypothesis, we took the task from the "dissimilar context" condition in Experiment 1 and used it as a testbed for learning arbitrary structural associations between events. In this variant of the base environment (see Asymmetrical Environment), the context cue for a maze during explore trials is a transformed version of the context cue during exploit trials. Note that we used a fixed transformation matrix across all mazes for simplicity in the current simulations, but theoretically it could be replaced with any complicated structures that need to be learned. We then took the base agent that retrieves the memory (value) that corresponds to the most similar query and key, and added a PFC module on top of the query and key generation process (Fig. 2). This added modulation allows extra flexibility in the expressiveness of the hippocampal encoding and retrieval processes, enabling the agent to recall memories given query-key matches between the current situation and all memory keys, as long as their is a learnable relationship between them. Results suggest that adding PFC top-down control over hippocampal episodic memory significantly improves the agent's ability to exploit an environment it has never seen before, by recalling memories that are structurally related to the current situation (Fig. 4). This PFC top-down modulation also outperforms a sensory-driven bottom-up memory retrieval condition, where the agent retrieves memories that share similar sensory inputs (including the context cue and its surroundings), and a random memory condition, where the agent retrieves a random memory from its values in the hippocampus.

## Experiment 3: PFC Learns Goal-Dependent Structures for Flexible Episodic Control

The results from Experiment 2 show that PFC can learn to top-down modulate hippocampal episodic memory during encoding and retrieval to learn an arbitrary structure between events. Building on this finding, we further extended the task to include multiple structures that need to be learned and flexibly represented by PFC modulatory activities for guiding truly goal-directed behaviors. Numerous experiments have demon-

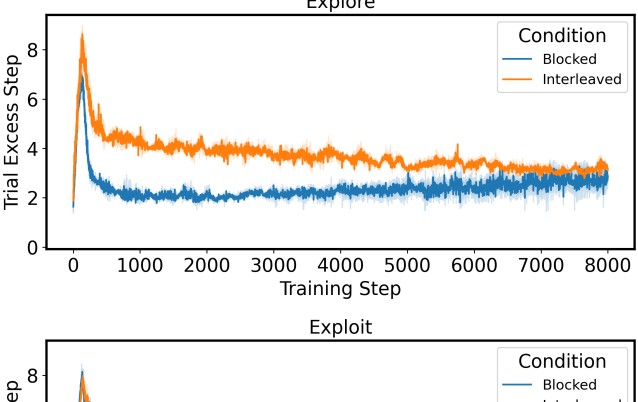

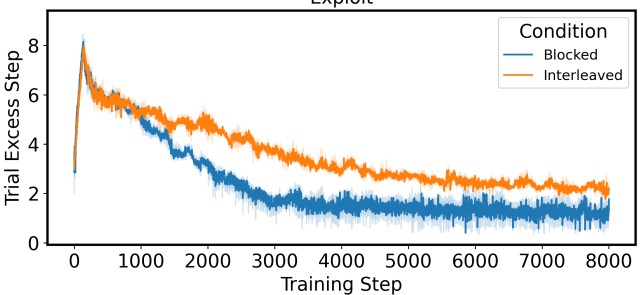

Figure 5: Experiment 3 results. Blocked training outperformed interleaved training in explore and exploit phases. Quickly learning episodic memory and constantly recalling it could hurt performance in exploration, as indicated by an initial quick drop in excess steps followed by eventual increases at the end of the run. NOTE: Shaded areas show SEM across runs.

strated the importance of PFC in decision making tasks that show active maintenance of goal representations for generalization (Miller & Cohen, 2001; Seger & Peterson, 2013; Baram et al., 2021; Samborska et al., 2022). On the other hand, the hippocampus often represents even-specific information (Marr et al., 1991; O'Reilly & McClelland, 1994; Yassa & Stark, 2011; Reagh & Ranganath, 2023) despite showing some goal representations (Crivelli-Decker et al., 2023; Brown et al., 2016). In this experiment, we ask whether the PFC can learn to flexibly switch between different query and key strategies to modulate hippocampal episodic memory depending on the current goal of the task, under what circumstances such capability can be learned, and what representations are formed in PFC and HPC when facing events that differ in goals.

We took the task from Experiment 2 and added a binary bit to represent the current goal in the observation space. The goal bit is randomly selected for each trial and is associated with two different hidden target locations that give rewards once the agent reaches them. Note that an agent with goal 1 cannot get a reward at the hidden target location associated with goal 2, which forces the agent to learn both structural relationships between explore and exploit trials to maximally utilize its episodic memory.

Our initial experiment doubled training time and interleaved explore trials for goals 1 and 2. However, the agent only partially learned to use episodic memory by the end of train-

ing, showing worse performance compared to Experiment 2 (Fig. 5). Representation analysis revealed that the agent successfully learned query-key paring only for goal 1, while pairings for goal 2 remained nearly random (Fig. 6). Inspired by human experiments and computational models on continual learning (Flesch et al., 2018; Park et al., 2020; Russin et al., 2022; Dekker et al., 2022), we improved learning of both strategies by introducing blocked learning of explore trials for goals 1 and 2. While this approach seems counterintuitive, as neural networks typically train better with interleaved learning to prevent catastrophic forgetting (McClelland et al., 1995), constantly switching between different tasks can create switch costs (Russin et al., 2022; Flesch et al., 2022) that result in noisy hidden state representations. In our case, working memory benefits from accumulating information about a particular goal in the maze across multiple trials in an episode, and storing such clean hidden representations in episodic memory better captures the agent's knowledge about navigating with that specific goal.

When trained with a blocked design during explore episodes, we observed learning curves that eventually matched the performance level of Experiment 2 (single-goal learning). Interestingly, we also observed a similar effect of episodic memory eventually hurting performance in exploration, as indicated by initial quick drop of the excess steps (due to cleaner working memory content) and eventual increases at the end of run (due to interference from episodic memories). Further analyses of the agent's learned representations showed successful query-key pairings for both goals, with queries and keys belonging to the same goal sharing similar representations. In other words, PFC learned to amplify the goal signal in the sensory input to better modulate hippocampal episodic memory when facing different goals in the same environment. Additionally, hippocampal representations showed event-specific patterns, where each event in a maze had a relatively unique representation, although some similarity remained between mazes that had different goals but were structurally related to the same context (Fig. 6). These representation patterns suggest that PFC-HPC interactions support goal-directed decision making in novel situations, with PFC exerting top-down control over HPC episodic memory to optimally serve the current task goal.

## Discussion

We introduced a computational model that integrates working memory with episodic memory via a PFC-HPC interaction framework. Our three experiments shed light on how the brain might achieve generalization, including far transfer, and suggest how these observations could contribute to improvement of future AI models: 1) When the current context is similar to past experiences, near transfer can be achieved by recalling the most similar memories in any feature space; 2) When current context is dissimilar to past experiences (or out-of-distribution), generalization is difficult and requires modeling the underlying structure to relate the current situation to rel-

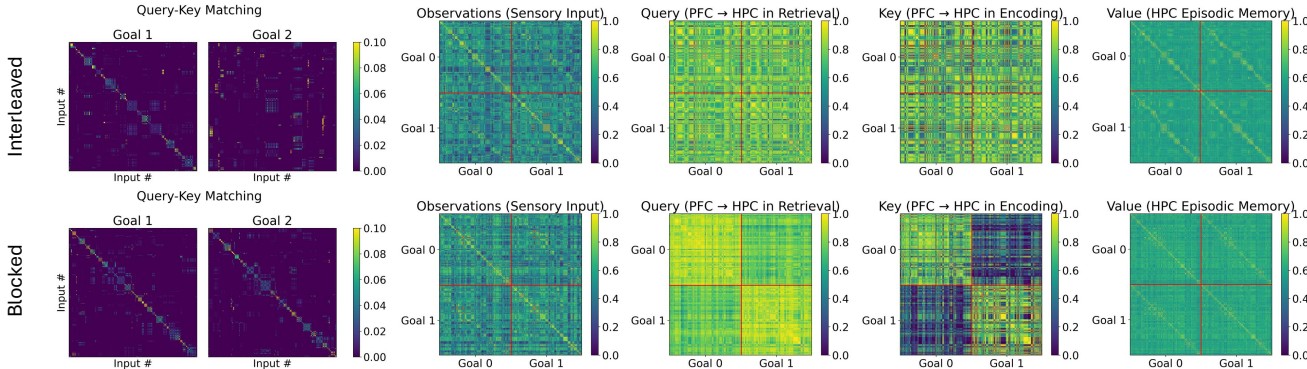

Figure 6: Representational analyses of blocked and interleaved agents in Experiment 3. Blocked training significantly improved the agent's ability to learn two underlying transformation matrices for two different goals, as indicated by better matching between queries and keys for the same events and higher within-goal similarities but lower between-goal similarities in both queries and keys. When these two structures were successfully learned, PFC amplified the goal bit in the sensory input to better guide HPC episodic control, essentially encrypting and deciphering HPC episodic memories in a PFC-specific subspace. The hippocampus represented event-specific information, with each event having a relatively unique representation, although some similarity remained between mazes that had different goals but were structurally related to the same context.

event memories. The PFC can learn many such structures depending on the current goal of the task; 3) When there are multiple structures to be learned, blocked training might provide advantages in learning each structure. This is evident in humans but rarely done in practical AI training due to catastrophic interference. We argue that learning and generalization may benefit from avoiding competing goal signals in working memory.

Generalization has been at the center of cognitive psychology and neuroscience research (Woodworth & Thorndike, 1901; Watson & Rayner, 1920; Tolman, 1948). Out of behaviors observed in generalization, far transfer is one of the most rare ones, resulting in debate about whether it truly exists (Barnett & Ceci, 2002). Despite efforts finding neural correlates of far transfer in the brain (Urbanski et al., 2016; Hobeika et al., 2016; Whitaker et al., 2018), it is yet to be described how the brain can computationally achieve far transfer. In particular, how does it represent the relational structure or rule (Taylor et al., 2021) that is shared between the current task and past experiences? Recent progress in AI, especially among Large Language Models (LLMs), provides an intriguingly elegant way to represent relationships between events (or sequences of tokens) – that is, abstracting all the semantic relationships between tokens into a high-dimensional space. As a result, the more training data and model parameters we have, the more statistical regularities can be picked up, resulting in somewhat emergent capabilities that can be transferred to solve daily tasks. However, the question remains how to effectively inject new semantic knowledge or episodic memories into such models, and how to form connections between them and the existing knowledge in the abstract parameter space.

Our model offers a potential mechanism for PFC modulation of hippocampal memory to support goal-directed decisions in novel situations, building on prior work on PFC–HPC

interactions. Prior work has emphasized the importance of top-down modulation from the PFC for context-sensitive retrieval (Chateau-Laurent & Alexandre, 2022), which shows that contextual signals from PFC can improve memory retrieval in a hippocampus-inspired architecture. Similarly, our model learns to control encoding and retrieval dynamically based on task structure. The Neural Episodic Control (Pritzel et al., 2017) laid groundwork for fast learning via key–value memory but lacked goal-modulated control. We extend this idea by showing how PFC modulation enables memory retrieval beyond surface similarity. More recently, the EGO model (Giallanza et al., 2024) provided a complementary computational account of episodic generalization, combining memory with contextual control. Both EGO and our model emphasize that effective generalization—particularly in far transfer scenarios—requires more than sensory matching; it depends on learning structured, goal-sensitive mappings between past and present contexts. While EGO models this via a latent control signal, we implement it through top-down query-key modulation from the PFC to the HPC. Together, these frameworks converge on a shared hypothesis: that structured generalization emerges from dynamic, goal-dependent interactions between hippocampus and prefrontal cortex. Promising future directions include incorporating replay-based consolidation mechanisms (Singh et al., 2022), adaptive storage via event boundary detection (Lu et al., 2022), and more biologically grounded hippocampal modules featuring pattern separation and completion (Zheng et al., 2022, 2024).

Our current simulations fall short in that the environment setup is still relatively abstract and lacks the complexity of real-world RL tasks as seen in robotics research. However, the principles we observed should scale up to more complex tasks requiring learning of more abstract relationships between current situations and past experiences. For example, one could

replace the fixed transformation matrices in the current experiments with arbitrarily complicated algorithms and study how to best learn to perform "algorithmic reasoning" using neural networks (Veličković & Blundell, 2021).

In summary, we show how PFC-HPC interactions enable flexible generalization through structured memory retrieval, with the PFC learning to modulate HPC episodic memory based on abstract relationships between tasks. These insights advance our understanding of how biological systems generalize to novel situations while suggesting new approaches for AI.

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

# Supplementary Materials

## Gating

While our main experiments used soft attention over episodic memory, which produces a non-zero weighted sum of values, we acknowledge this implementation may oversimplify hippocampal retrieval. Sparse, context-sensitive retrieval that depends on gating mechanisms (e.g., basal ganglia) is more biologically plausible.

In follow-up experiments, we explored adding a learned gating mechanism that mimics basal ganglia modulation over episodic memory retrieval. Specifically, a small policy network outputs a scalar gating value conditioned on the current PFC state, which multiplies the retrieved episodic memory vector before it is integrated via self-attention. This enables the agent to suppress memory retrieval when query-key similarity is low or memory retrieval is expected to be irrelevant or harmful (e.g., during exploration in novel mazes). This gating mechanism could potentially reduce performance interference in exploration and lead to more context-appropriate use of episodic memory. We plan to incorporate this extension in future versions of the model.

## EC - HPC - PFC and value representation

The architecture in Figure 2B draws inspiration from known anatomical pathways between entorhinal cortex (EC), hippocampus (HPC), and prefrontal cortex (PFC), but it also makes simplifications to focus on the computational mechanism of top-down episodic control.

In our model:

- **EC** represents bottom-up, sensory-driven retrieval mechanisms, where queries to memory are generated directly from current input without task-modulated control.

- **PFC** generates top-down, goal-modulated queries and keys, enabling selective retrieval of memories based on abstract structure rather than surface features.

The **value** stored in memory corresponds to the hidden state of the PFC's reservoir network at the time of encoding. This represents a compressed snapshot of the agent's internal belief and goal-relevant information at that moment. While this simplification helps isolate the role of PFC in memory control, we acknowledge that in biological systems, hippocampal values would likely also incorporate EC-derived sensory inputs and broader cortical states.

Thus, our current implementation frames HPC as storing PFC working memory representations, with EC pathways serving as a contrasting retrieval route. Future extensions could explore more biologically grounded representations of episodic content that combine both PFC and EC contributions.

## Model and Brain Correspondence

In our model, the network connections do not strictly correspond to anatomically defined pathways in the brain. Some connections may represent combinations of multiple biological pathways, which could be unidirectional or bidirectional—for example, the connections between the prefrontal cortex (PFC), hippocampus (HPC), and entorhinal cortex (EC). In addition, EC in the diagram is not directly implemented in the model and is for demonstrating the idea that EC is a relay station of sensory input into the HPC. Computationally, we treat the PFC as modulating both the query, which prompts the HPC to retrieve relevant memories, and the key, which serves as an index into the hippocampal memory store.

For simplicity, we represent the memory values as the PFC's internal hidden states, since these are directly relevant to solving the task. However, this is not a hard constraint. The model can be extended to store richer memory content that includes activity from other regions if doing so proves useful for more complex tasks or closer biological fidelity.

To further clarify our interpretation, Figure 7 illustrates the roles of different brain regions that may correspond to memory encoding and retrieval in our model. During encoding, the HPC stores key–value pairs, where the key acts as a neural index and the value represents the memory content. Keys can be formed either through bottom-up sensory pathways (e.g., via EC) or through top-down modulation from PFC. Similarly, during retrieval, a query is sent to the HPC to retrieve a weighted combination of stored memory values, depending on query–key similarity. This query can again originate bottom-up (from EC) or be modulated by PFC in a top-down manner. The final output is a soft retrieval—a weighted sum over values, allowing partial activation of multiple memories.

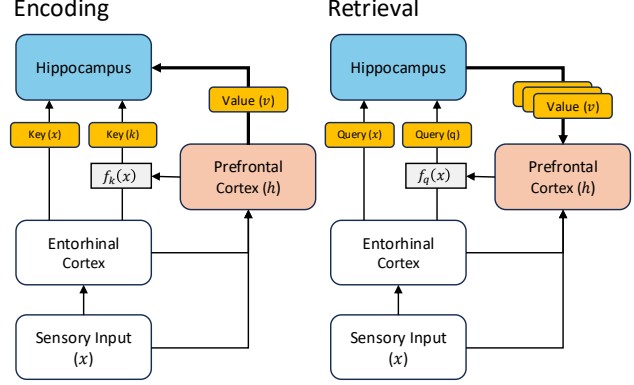

Figure 7: Diagram of encoding and retrieval. During encoding, the hippocampus stores key–value pairs, where the key serves as a neural index for accessing stored content. Keys can be computed either bottom-up (from the entorhinal cortex) or top-down (from the entorhinal cortex modulated by the prefrontal cortex). During retrieval, a query is sent to the hippocampus to retrieve a weighted combination of memory values, based on the similarity between the query and each stored key. As with encoding, queries can originate bottom-up or be shaped by top-down signals. The final retrieved memory is a softmax combination of values.

