# OpenReview forum: "Flexible Prefrontal Control over Hippocampal Episodic Memory for Goal-Directed Generalization"
_ccneuro.org/CCN/2025/Proceedings — CCN 2025 Proceedings asProceedingsPoster_

### Official Review · Reviewer_gRxA · 2025-03-10
**innovative ideas and impressive, yet understandable, findings; however, additional details regarding the implementation would be appreciated**

**Soundness:** 3
**Clarity:** 3

**Comments:**

This paper presents computational models that bridge neuroscience and reinforcement learning to explain how the brain generalizes from episodic memory to support goal-directed behavior in novel contexts. Using a query-key-value architecture, the authors model how the hippocampus stores context-specific representations and leverages relevant past experiences through episodic memory retrieval, while top-down modulation from the prefrontal cortex (PFC) enables generalization beyond mere sensory similarity.

The paper addresses critical questions about how the brain achieves generalization across diverse contexts and how its neural mechanisms can be harnessed to improve transfer learning in neural networks. The authors effectively define the problem by integrating insights from previous work.

One of the study’s strengths is its progressive experimental design, which adds complexity to allow neural network agents to generalize at more abstract, conceptual levels. However, due to the constraints of the conference proceeding format, some details of the neural network implementation and performance estimation remain unclear. Additional results—such as analyses of hidden layer representations, how the PFC control was implemented to provide flexibility, or behavioral performance changes corresponding to variations in the distribution of dot similarity—would provide deeper insights into the dynamics at play. Nonetheless, it is understandable that space limitations preclude a more exhaustive presentation, and these issues could be further addressed during the CCN presentation.

I believe that this study not only addresses timely and broadly relevant topics but also proposes an innovative approach supported by sound evidence. The insights provided have the potential to spark significant discussions and contribute to future research in AI, cognitive science, and neuroscience. Its interdisciplinary nature and rigorous experimental design offer valuable contributions that can shape the direction of subsequent studies in these fields.

**Expertise:**

3

**Interest:**

3

---

> ### Author Rebuttal · Authors · 2025-04-15
>
> We’re grateful for the reviewer’s enthusiastic assessment and helpful suggestions.
>
> We acknowledge that the current presentation omits some implementation and representational details due to space limits. We also appreciate the suggestion to connect more deeply with broader themes in neuroscience and AI, which we will integrate more explicitly in the conclusion.

---

### Official Review · Reviewer_72tU · 2025-03-23
**comments**

**Soundness:** 3
**Clarity:** 2

**Comments:**

The authors proposed a reinforcement learning model for goal-directed generalisation. Specifically, they introduced a coupled system mimicking interactions between the prefrontal cortex (PFC) and hippocampus (HPC). In this system, the PFC learns to generate query-key representations to encode and retrieve goal-relevant episodic memories and modulates HPC memories top-down based on current task demands. The authors demonstrated their model through three simulations and showed that: 1) combining working memory with selectively retrieved episodic memory enhances decision-making in similar environments; 2) top-down control from the PFC over the HPC improves the learning of arbitrary structural associations between events, facilitating generalisation to novel environments; and 3) the PFC encodes similar representations during both the encoding and retrieval of goal-relevant memories, whereas the HPC exhibits event-specific representations. Overall, this is a strong paper addressing important questions regarding hippocampal episodic memory and its role in decision-making in novel situations.

However, I have a few concerns:

1, I found it difficult to fully understand the methods section without first understanding its relevance to the results. For instance, the introduction of the asymmetrical environment would benefit from more explanation regarding its purpose. I only fully grasped the significance of this after reading the results section.

2, While this is a valuable model, it is relatively simple. Are there any predictions that could be made through this model that might be tested in rodent or human data? Additionally, in the discussion, the authors mention that the principles outlined should scale to more complex situations. Could the authors elaborate on this point?

3, The authors should discuss similar models and provide comparisons with their own model in the discussion.

4, What is the function of the modulation signal m? The text states it is generated by a neural network, but Eq. 8 appears to be only a non-linear transformation. This needs clarification.

5, In Fig. 2b, is there a specific reason that the entorhinal cortex (EC) provides the query and key to the HPC, while the HPC sends the value back to the EC? Additionally, since the value is the hidden state of the reservoir network, which is intended to model the PFC, why is this value sent from the HPC to the EC? Is there evidence that the PFC sends value information to the HPC for storage? Please clarify this point.

6, Perhaps I missed it, but why do we observe sudden increases in excess steps in Figs. 3, 4, and 5 at the beginning of training?

**Expertise:**

2

**Interest:**

1

---

> ### Author Rebuttal · Authors · 2025-04-15
>
> Thank you for the thoughtful review on both strengths and areas for clarification.
>
> 1. Motivation for asymmetric environment
>
> We agree the motivation was underdeveloped. We will revise the methods to clarify that asymmetry was introduced to test generalization beyond surface similarity, requiring PFC to learn abstract mappings.
>
> 2. Predictions for biological systems and scaling complexity
> Yes—our model makes testable predictions. For example:
>
> - When two tasks share a structural relationship but differ in surface features, successful transfer should depend on intact PFC function.
>
> - If a subject learns two goals that share the same context cues but different latent task structures, blocked training should facilitate learning, and PFC activity should reflect goal modulation.
>
> - HPC representations of retrieved memories will differ depending on the goal or task context, even for the same stimuli.
>
> Regarding complexity scaling: by saying “principles should be general enough to scale…”, we meant that the fixed transformation matrices in Exp. 2/3 could in principle be replaced with more complex functions, which PFC-HPC systems (or neural nets) could learn given sufficient training.
>
> 3. Comparison to similar models
>
> We will add comparisons to NEC, hippocampal RL models, and recent PFC-HPC models, clarifying how our use of episodic memory differs.
>
> 4. Clarification of modulation signal m
>
> Eq. 8 describes a non-linear transformation of the output of the network f_filter, which maps a fixed bias vector to a modulation signal. The non-linear transformation gives the modulation signal upper and lower bounds. The whole function of m is to select relevant parts of input information for going into the reservoir RNN. We have added text to further clarify that and provided citations for the usage of m.
>
> 5. EC ↔ HPC ↔ PFC and value representation
>
> The diagram was inspired by known anatomy, but the functional mapping is simplified.
> EC = bottom-up retrieval from sensory input.
> PFC = top-down retrieval modulated by task demands.
> Memory values = RNN hidden state at encoding, i.e., working memory snapshot.
> Though simplified, this focuses on goal-relevant representations; future models could incorporate richer EC-derived content. We’ll clarify this in the Supplementary Materials.
>
> 6. Early spikes in excess steps
>
> The reviewer is right that these are due to early training instability, not memory interference, and are observed often in reinforcement learning experiments.

---

> > ### Comment · Reviewer_72tU · 2025-04-21
> >
> > Thank you for the clarification. I will maintain my score. Looking forward to it.

---

> > > ### Author Response · Authors · 2025-04-21
> > >
> > > I appreciate your engagement and look forward to the next steps.

---

### Official Review · Reviewer_Souu · 2025-03-28
**Flexible prefrontal control over hippocampal episodic memory for goal-directed generalization**

**Soundness:** 2
**Clarity:** 2

**Comments:**

Really interesting and timely paper. It proposes a model where the PFC learns to bias episodic memory retrieval in the hippocampus depending on task goals and structure. It captures key cognitive phenomena like proactive interference and switch costs, and provides a compelling case for the importance of top-down control in generalization. The setup is a bit artificial but well thought-out, and the experiments are clear and informative.

This is very relevant for the CCN community, especially for people interested in PFC-HPC interactions, memory-guided behavior, and bridging neuroscience with AI. The idea that the PFC can modulate episodic queries to support far transfer touches on a central question in both fields. Could be of interest to a broad audience.

The model is coherent and well-supported by the experiments. Using a reservoir for the PFC is an interesting choice — would be good to clarify how hyperparameters were chosen (spectral radius, leak rate, input scaling). Also, the idea of having PFC modulate queries and keys (f_q, f_k) is powerful but a bit under-explained. There’s room to connect more with related models, like Opportunistic PFC (Chateau-Laurent & Alexandre, 2022), or Pilly et al., 2018.

Very clear writing, well-organized figures, and well-structured experiments. The three experiments each bring something new and build on one another nicely. The architecture is a bit dense but still understandable, and the diagrams help. The fact that the two architecture figures do not seem to show the same connections is strange though.

The paper is related to the idea of a “construction system” (Hassabis & Maguire, 2009). It would be good to tie this more explicitly to the broader literature on planning and imagination. It also echoes work in hippocampal RL like NEC (Pritzel et al.), and connecting those dots might strengthen the paper’s relevance. One nitpick: the note in the caption of Fig 4 is written twice.

Overall, this is a solid and exciting contribution. It bridges neuroscience and AI in a meaningful way and provides a clear computational role for PFC-HPC interactions.

**Expertise:**

3

**Interest:**

3

---

> ### Author Rebuttal · Authors · 2025-04-15
>
> We appreciate the reviewer’s positive remarks and helpful suggestions to strengthen the paper.
>
> 1. Reservoir hyperparameters
>
> The spectral radius, leak rate, and input scaling were selected based on prior work on reservoir computing for memory tasks (e.g., McKee, 2024) and not tuned extensively. We will clarify these choices in the revision and cite the relevant literature.
>
> 2. Query/key modulation (f_q, f_k)
>
> In Experiments 2 and 3, f_q and f_k are implemented as independent learned MLPs. f_q generates top-down queries from PFC to the HPC, while f_k encodes keys during memory storage. They do not share weights, as they serve distinct functional roles. We will expand on this and improve clarity around their interpretation and implementation.
>
> 3. Related models and theoretical framing
>
> We agree that connecting to related work (e.g., Opportunistic PFC, NEC, construction system) will strengthen the impact. We will revise the discussion to situate our model more clearly within these frameworks and emphasize the contribution of flexible episodic control for generalization.
>
> 4. Architectural inconsistencies and Figure 4 caption
>
> Thanks for pointing this out—we will revise the architecture figures to ensure consistency, and correct the duplicated note in Figure 4’s caption.

---

> > ### Comment · Reviewer_Souu · 2025-04-18
> >
> > _(This is TPC providing a pointer to the [**Official Comment**](https://openreview.net/forum?id=7hhz5ToJnM&noteId=YqKF22D5PF) from Reviewer Souu starting with:_
> >
> > > Thank you for the thorough  ...
> >
> > _With this **Rebuttal Comment** posted on behalf of the Reviewer, the Authors can respond with one final **Reply Rebuttal Comment** during the Author-Reviewer Discussion phase.)_

---

> > > ### Author Response · Authors · 2025-04-21
> > >
> > > Thank you for the thoughtful feedback and for taking the time to engage deeply with our work!

---

### Official Review · Reviewer_mg59 · 2025-03-31
**Interesting modeling work that characterize the interaction between working memory and episodic memory in goal-directed actions**

**Soundness:** 2
**Clarity:** 2

**Comments:**

Overall very interesting work. I have two major comments about this paper.
The formulation of retrieved memory seems to suggest that regardless of the similarity between query and key, there is always information from episodic memory integrated with working memory via attention. I don't think this is the best way to characterize this process given the sparse representation of memory episodes in hpc. A query does not always guarantee successful retrieval and the retrieved memory might not be a weighted sum.
Exp 3 showed that blocked training is significantly better than interleaved training when looking at the query-key mapping. I'm wondering if this is due to the way goal is represented in the model such that the working memory network cannot perform efficient maintenance and gating of goal status.

**Expertise:**

2

**Interest:**

2

---

> ### Author Rebuttal · Authors · 2025-04-15
>
> We thank the reviewer for the insightful and constructive feedback.
>
> 1. On always retrieving memory and query-key similarity
>
> We agree with the reviewer that retrieval should not always succeed, especially given the sparsity of episodic representations in the hippocampus. In our current implementation, retrieval uses a softmax over stored values, which indeed yields a non-zero weighted sum. However, we have explored versions of the model with gating mechanisms inspired by basal ganglia control, where retrieval is controlled by a learned network policy, which outputs a single scalar for memory retrieval. These results were omitted due to space but will be referenced in the text and discussed in more detail in the revision under Supplementary Materials due to length limit.
>
> 2. On blocked vs. interleaved training and goal representation
>
> The reviewer raises an important point about working memory maintenance of goals. Indeed, blocking gives the reservoir RNN a chance to get a cleaner version of representation compared to interleaving, which accumulates information about irrelevant information (another goal) when being stored. As a result, our representational similarity analyses confirmed that goal signals are more robustly represented and separable under blocked training compared to interleaved.

---

### Meta-Review · Area_Chair_i3Xc · 2025-05-06

**Ccn Recommendation:** Accept as Proceedings

**Metareview:**

There was a consensus among reviewers that the paper is methodologically sound and makes a meaningful contribution by integrating PFC-HPC interactions into a reinforcement learning framework to account for goal-directed generalization. The main questions centered on model assumptions and requests for clarifying technical details, which the authors addressed satisfactorily in their revision.  While the reviewers differed in their assessment of the paper’s scope—with two self-identified experts emphasizing its broad interdisciplinary relevance, and two others viewing it as more specialized—the AC agrees that the work holds broad interest for the CCN community. Taken together, the overall review were positive and the reviewers’ concerns were adequately addressed,  the AC is happy to recommend acceptance.

**Summary:**

Two reviewers highlighted the paper’s broad interdisciplinary relevance, while the other two considered its appeal more disciplinary or specialized. In terms of soundness, two reviewers rated it as strong and two as adequate. Most found the paper adequately clear, with one reviewer rating its clarity as exceptional.

One reviewer raised concerns about the assumption that memory retrieval always succeeds based on query-key similarity, noting that this may not accurately reflect the sparse nature of episodic memory representations in the hippocampus. In response, the authors clarified that they had also explored alternative models inspired by basal ganglia control, in which retrieval is governed by a learned network policy that outputs a single scalar signal. Additional reviewer comments requested further clarification on technical details—such as neural network implementation, choice of hyperparameters, behavioral performances,  rationale for the asymmetrical task environment, the mapping among key brain regions, and comparisons with related models. The authors addressed these points thoroughly within the manuscript’s space limitations, providing relevant explanations and revisions.

**Expertise:**

2